# A Review of Deep Learning Methods for Compressed Sensing Image Reconstruction and Its Medical Applications

**Yutong Xie** [1] 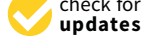 **and Quanzheng Li** [2,3,4,*]

1    Academy for Advanced Interdisciplinary Studies, Peking University, Beijing 100871, China; xieyutong@pku.edu.cn
2    MGH/BWH Center for Clinical Data Science, Department of Radiology, Massachusetts General Hospital and Harvard Medical School, Boston, MA 02114, USA
3    Center for Advanced Medical Computing and Analysis, Department of Radiology, Massachusetts General Hospital and Harvard Medical School, Boston, MA 02114, USA
4    Gordon Center for Medical Imaging, Department of Radiology, Massachusetts General Hospital and Harvard Medical School, Boston, MA 02114, USA
*    Correspondence: li.quanzheng@mgh.harvard.edu

**Abstract:** Compressed sensing (CS) and its medical applications are active areas of research. In this paper, we review recent works using deep learning method to solve CS problem for images or medical imaging reconstruction including computed tomography (CT), magnetic resonance imaging (MRI) and positron-emission tomography (PET). We propose a novel framework to unify traditional iterative algorithms and deep learning approaches. In short, we define two projection operators toward image prior and data consistency, respectively, and any reconstruction algorithm can be decomposed to the two parts. Though deep learning methods can be divided into several categories, they all satisfies the framework. We built the relationship between different reconstruction methods of deep learning, and connect them to traditional methods through the proposed framework. It also indicates that the key to solve CS problem and its medical applications is how to depict the image prior. Based on the framework, we analyze the current deep learning methods and point out some important directions of research in the future.

**Keywords:** compressed sensing; magnetic resonance imaging; computed tomography; positron emission tomography; deep learning

## 1. Introduction

Compressed sensing (CS) is an important problem in signal process. It can be described as reconstructing signal $\mathbf{x}$ from its measurement $\mathbf{y}$ where $\mathbf{x} \in \mathbb{R}^n$, $\mathbf{y} \in \mathbb{R}^m$, $m < n$ and $\mathbf{y}$ is obtained in the following form:

$$\mathbf{y} = \mathbf{A}\mathbf{x} + \boldsymbol{\varepsilon}. \tag{1}$$

$\mathbf{A} \in \mathbb{R}^{m \times n}$ defines the measuring system and $\boldsymbol{\varepsilon}$ is the noise. Reconstructing high quality images or signals has been an active area of research and holds high value in many applications, especially in medical imaging reconstruction such as computed tomography (CT), magnetic resonance imaging (MRI) and positron-emission tomography (PET). In the past two decades, traditional CS theory has been established to reconstruct $\mathbf{x}$ from $\mathbf{y}$. Due to $m < n$, solving the inverse problem is not easy. Based on sparsity of $\mathbf{x}$, many optimization algorithms were proposed to solve it. Though the traditional CS theory is pretty and elegant, there are still some drawbacks. For example, classic algorithms usually take a long time to solve the CS problem. Recently, deep learning—a data driven method—has demonstrated tremendous success in many fields and there is a trend to use it to solve the CS problem. Deep learning is a class of machine learning approaches that utilize cascaded layers of linear and nonlinear functions to learn the complex mapping from data. When networks go deeper with more parameters, its capability of learning features is improved, which allows the deep network to learn complex functions directly from data

without human-crafted features. The core of deep learning, deep neural network, dates back to 1950s. Modern techniques, including improvements on optimization algorithm (stachastic gradient descent (SGD), rectified linear units (ReLU), batch normalization, dropout, shortcut connection et al.), more effective network architectures (convolutional neural networks (CNN), recurrent neural networks (RNN), generative adversarial networks (GAN)), the availability of large datasets and stronger computational power of hardware (GPU and parallel computing), contribute to the tramendous success of deep learning. In this review, we focus on the application of deep learning in the general CS problem and three types of medical imaging—CT, MRI and PET.

Different from some other reviews [1] which divide deep learning methods into several categories, we attempt to construct a unified framework to cover all these categories. The analysis begins with the variational model and a simple algorithm. Usually, the object of a variational model is to minimize the following function:

$$\min_{\mathbf{x}} f(\mathbf{y}, \mathbf{A}\mathbf{x}) + \sum_{i=1}^{K} \lambda_i R_i(\mathbf{x}). \tag{2}$$

$f(\mathbf{y}, \mathbf{A}\mathbf{x})$ represents the data consistency and $R_i$ $(i = 1, \ldots, K)$ are regularization terms. For simplicity, suppose that there is only one regularization term and $f(\mathbf{y}, \mathbf{A}\mathbf{x}) = \|\mathbf{y} - \mathbf{A}\mathbf{x}\|_2^2$, then Equation (2) can be written as follows:

$$\min_{\mathbf{x}} \|\mathbf{y} - \mathbf{A}\mathbf{x}\|_2^2 + \lambda R(\mathbf{x}). \tag{3}$$

The common choice for $R(\mathbf{x})$ is Total Variation [2] or $\|\mathbf{W}\mathbf{x}\|_1$ where $\mathbf{W}$ is some linear transform such as the wavelet transform. We use a simple iterative algorithm to solve Equation (3). The iterative process can be written as follows:

$$\begin{cases} \mathbf{x}^{(k+\frac{1}{2})} = \arg\min_{\mathbf{x}} \|\mathbf{y} - \mathbf{A}\mathbf{x}\|_2^2 + \left\|\mathbf{x} - \mathbf{x}^{(k)}\right\|_2^2, \\ \mathbf{x}^{(k+1)} = \arg\min_{\mathbf{x}} \lambda R(\mathbf{x}) + \left\|\mathbf{x} - \mathbf{x}^{(k+\frac{1}{2})}\right\|_2^2. \end{cases} \tag{4}$$

This algorithm contains two steps. By geometric analysis, the first step moves $\mathbf{x}$ to a position closer to the hyperplane $\mathbf{y} = \mathbf{A}\mathbf{x}$ and the second step moves $\mathbf{x}$ to a position with lower value of $R(\mathbf{x})$. If we regard the regularization term as a depiction of the manifold of signals, the iterative algorithm derives a solution by alternatively moving $\mathbf{x}$ to the hyperplane and the manifold. The movement of $\mathbf{x}$ is shown in Figure 1.

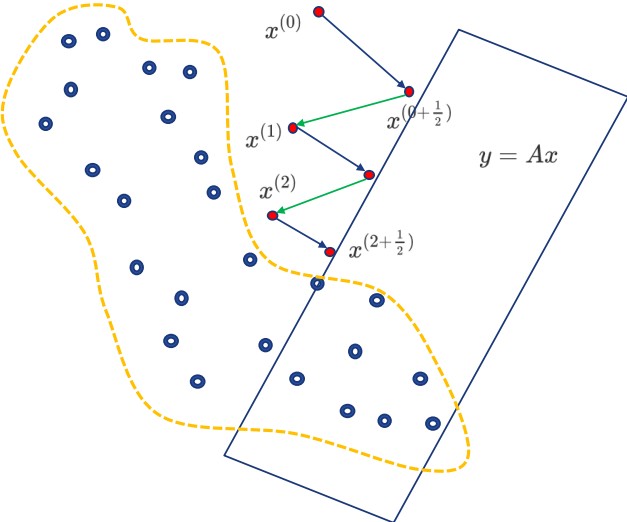

**Figure 1.** A simple algorithm to solve the variational model. Blue arrows stand for the first step of iteration and the green arrows for the second one.

From the Bayesian view, we can understand the role of regularization terms more clearly and figure out what the solving algorithm do. Suppose $\varepsilon \sim N\left(\mathbf{0}, \sigma^2 \mathbf{I}\right)$ and the prior distribution of $\mathbf{x}$ is $p(\mathbf{x})$. Then we derive the logarithmic posterior probability of $\mathbf{x}$ as follows:

$$\log p(\mathbf{x} \mid \mathbf{y}) = -\|\mathbf{y} - \mathbf{A}\mathbf{x}\|_2^2 + \lambda \log p(\mathbf{x}). \tag{5}$$

Here, for simplicity, the coefficient of $\|\mathbf{y} - \mathbf{A}\mathbf{x}\|_2^2$ is neglected. If we apply a simple first-order gradient method to maximize the posterior probability, the iteration will be in the following form:

$$\mathbf{x}^{(k+1)} = \mathbf{x}^{(k)} + \eta \mathbf{A}^H \left(\mathbf{y} - \mathbf{A}\mathbf{x}^{(k)}\right) + \eta \lambda \nabla \log p\left(\mathbf{x}^{(k)}\right) \tag{6}$$

where $\mathbf{A}^H$ is the conjugate transposition matrix of $\mathbf{A}$ and $\eta$ is the step length. It is easy to verify that $\eta \mathbf{A}^H \left(\mathbf{y} - \mathbf{A}\mathbf{x}^{(k)}\right)$ represents a direction toward the data consistency hyperplane $\mathbf{y} = \mathbf{A}\mathbf{x}$ and $\eta \lambda \nabla \log p\left(\mathbf{x}^{(k)}\right)$ toward higher prior probability. The geometric interpretation is illustrated in Figure 2. We can see the similarity between the variational model and the Bayesian model. In other words, regularization terms correspond to the representation of logarithmic prior distribution of $\mathbf{x}$. Thus, we have the following conjecture: the solving algorithm of the CS problem is to search a solution that is in the intersection of the data consistency and the prior information. It contains two parts. One is a "projection" operator to the image prior and the other one is to the data consistency.

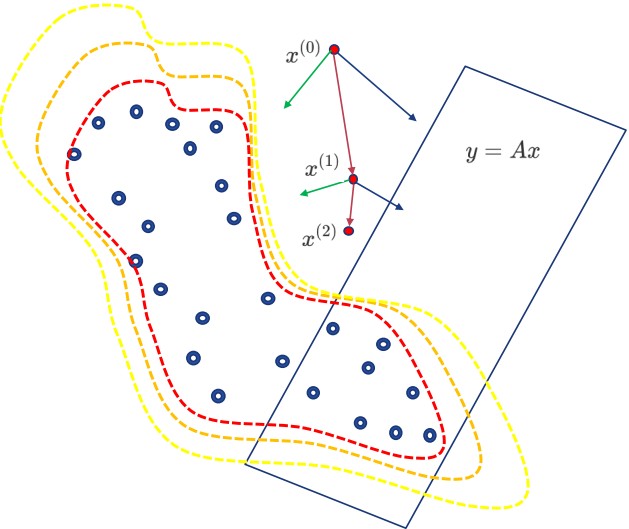

**Figure 2.** A simple algorithm to solve the Bayesian model. Two directions are denoted by blue and green arrows. The dash lines of different colors are used to represent the area of different prior probability. The probability value is from low to high when the color is changed from yellow to red.

Based on the geometric analysis of optimization algorithms, we can define a unified framework for solving the CS problem. Since the typical signals in CS are images and three applications reviewed here are medical imaging, we only discuss image signals through the review. Let $M_{\text{image}}$ be the manifold representing the image prior and $M_{\text{dc}}$ be the solution space of the data consistency. We define $P_{\text{image}}$ as a transform that projects $\mathbf{x}$ toward $M_{\text{image}}$ and $P_{\text{dc}}$ as one projecting $\mathbf{x}$ toward $M_{\text{dc}}$. We claim that a solving algorithm satisfies the framework $\mathcal{F}$ if it is composed of $P_{\text{image}}$s and $P_{\text{dc}}$s. Sometimes, $P_{\text{image}}$ can be further decomposed into three transforms:

$$P_{\text{image}} = \mathcal{V}_x \circ \mathcal{P}_x \circ \mathcal{U}_x, \tag{7}$$

where $\mathcal{U}_x : \mathbb{R}^n \to S, \mathcal{P}_x : S \to S, \mathcal{V}_x : S \to \mathbb{R}^n$. $\mathcal{U}_x$ transforms an image to a defined space $S$; $\mathcal{P}_x$ defines the "projection" operator in $S$ and $\mathcal{V}_x$ transforms the result back to the image space. This decomposition means that the image prior can be depicted in space $S$ rather than $\mathbb{R}^n$. For example, suppose the regularization term in the variational model is $\|\mathbf{W}\mathbf{x}\|_1$ where $\mathbf{W}$ is a wavelet decomposition operator. Then $\mathcal{U}_x$ is the wavelet transform, $\mathcal{V}_x$ is the inverse transform and $\mathcal{P}_x$ can be the soft-thresholding function. We find that the solving algorithms for the variational model or the Bayesian model satisfy $\mathcal{F}$. Though $\mathcal{F}$ is very simple, it is surprising that almost all deep learning methods solving the CS problem or its applications also satisfy this framework. Thus, the framework $\mathcal{F}$ provides a perspective to analysing solving algorithms. In addition, since CS belongs to inverse problems, this framework can be expanded to other inverse problems such deblurring, inpainting, etc. as long as we choose a feasible $P_{\text{dc}}$.

Our main contributions are as follows:

- We proposed a framework which unifies traditional iterative algorithms and deep learning approaches for CS reconstruction and its medical applications.
- We reviewed many works on reconstruction of CS, CT, MRI and PET, and analyzed them based on the proposed framework.
- Through the proposed framework, we built relationship between different reconstruction methods of deep learning and indicated that the key to solve CS problem and its medical applications is how to depict the image prior.

In later sections, we also divide deep learning methods into different categories. Nevertheless, the emphasis is to illustrate how these categories match the framework $\mathcal{F}$. This review is organized as follows. Section 2 describes deep learning methods used in general CS. Some works for CT reconstruction are reviewed in Section 3. Section 4 surveys recent deep learning methods for MRI reconstruction. Then, we provide some deep learning approaches for PET reconstruction in Section 5. Finally, we compare these methods and discuss future directions in Section 6 and concludes the review in Section 7.

## 2. Deep Learning Methods for Compressed Sensing

### 2.1. Overview

In the general CS reconstruction problem, usually $\mathbf{x}$ is a natural image and $\mathbf{A}$ is a Gaussian random matrix. In this section, we divide deep learning approaches into five categories and analyze how each one matches framework $\mathcal{F}$.

### 2.2. Model-Based Methods with Learnable Parts

The first category is model-based methods with learnable parameters. These methods may be traced back to learned iterative shrinkage and thresholding algorithm (LISTA) [3]. Convolutional Neural Networks (CNN) or other neural networks are not used. Instead, some pre-fixed parameters or functions in traditional algorithms are learned from training data. Generally speaking, through loss function and back-propagation method, these algorithms can be regarded as a trainable network [4–7]. Suppose the traditional algorithm is $\text{Alg}(\cdot; \theta)$ where $\theta$ is the pre-fix parameters. Then the reconstruction of it can be written as $\text{Alg}(\mathbf{y}; \theta)$ where $\mathbf{y}$ is the measurement. Let $(\mathbf{x}_i, \mathbf{y}_i)_{i=1}^N$ is the training set and $L$ is the loss function, then the training process is to optimize over $\theta$ by $\min_\theta \sum_{i=1}^N \frac{1}{N} L(\text{Alg}(\mathbf{y}_i; \theta), \mathbf{x}_i)$ where $\mathbf{x}_i$ is the training label. Therefore, $\text{Alg}(\mathbf{y}_i; \theta)$ can be regarded as a trainable network with learnable parameters $\theta$. The purpose of applying data-driven scheme is diverse. Some are to reduce computation cost, some to ascertain best parameters and some to make regularization terms closer to the image prior. Since the overall form is not changed and the original algorithm itself satisfies the framework $\mathcal{F}$, these methods still consist of $P_{\text{image}}$s and $P_{\text{dc}}$s and therefore match the framework $\mathcal{F}$.

The authors of [4] proposed to replace the soft thresholding function in iterative shrinkage and thresholding algorithm (ISTA) by other learnable non-linear functions. They used cubic spline functions as basis functions and learned the weights $c_k$. The alternative function has the following form:

$$\varphi(z) \triangleq \sum_{k=-K}^{K} c_k \psi\left(\frac{z}{\Delta} - k\right), \tag{8}$$

where $\psi$ is the cubic spline function, $\Delta$ is the granularity parameter and $K$ is the number of basis functions. Given fixed $T$ iterations, L2 norm loss between final reconstruction results and real images was used to train the weights $c_k$. In addition, the authors of [5] not only used Equation (8) but also trained the step length. Similarly, Gaussian kernel functions were used as basis functions to replace the proximal operator in ISTA [8]. The shrinkage function $\psi^t(u)$ is written as follows:

$$\psi^t(u) = \sum_{k=1}^{K} c_k^t \phi_k(u), \text{ where } \phi_k(u) = ue^{-\frac{(k-1)u^2}{2\tau^2}}. \tag{9}$$

$\phi_k$ is the Gaussian kernel function and $K$ is the number of basis functions and $t$ is used to represent different steps. To reduce number of learnable parameters, the authors of [6] proposed to employ linear expansion of thresholds (LET) to substitute the soft thresholding function. Besides, they considered fast ISTA (FISTA) instead of ISTA.

The authors of [7] proposed a novel network, ISTA-Net, which replace the linear transform in the regularization term by a two-layer neural network. In original algorithm, the second step of iteration is a proximal operator which has the following form:

$$\text{prox}_{\lambda\phi}(\mathbf{r}) = \mathbf{W}^\top \text{soft}(\mathbf{Wr}, \lambda). \tag{10}$$

They used $F(\cdot)$ and $\widetilde{F}(\cdot)$ (two-layer neural networks) to substitute $\mathbf{W}$ and $\mathbf{W}^\top$. Since $\mathbf{W}^\top\mathbf{W} = \mathbf{I}$, the constraint of $\widetilde{F} \circ F = \mathbf{I}$ is added to the loss function. It has the following form:

$$\begin{aligned}
\mathcal{L}_{\text{total}}(\Theta) &= \mathcal{L}_{\text{discrepancy}} + \gamma\mathcal{L}_{\text{constraint}} \\
&= \frac{1}{N_b}\sum_{i=1}^{N_b}\left\|\mathbf{x}_i^{(N_p)} - \mathbf{x}_i\right\|_2^2 + \frac{1}{N_b}\sum_{i=1}^{N_b}\sum_{k=1}^{N_p}\left\|F^{(k)}\left(F^{(k)}(\mathbf{x}_i)\right) - \mathbf{x}_i\right\|_2^2,
\end{aligned} \tag{11}$$

where $N_b$ is the amount of data and $N_p$ is the number of iterations. The input of the network is an initial reconstructed image. Based on ISTA-Net, they considered residual learning and proposed a modified version ISTA-Net+.

A recent work [9] proposed to substitute the convolutional operator in transform learning algorithm by a learnable convolutional layer with $3 \times 3$ kernels. The object function is as follows:

$$\min_{\mathbf{x}, \boldsymbol{\alpha}_k} \|\mathbf{y} - \boldsymbol{\Phi}\mathbf{x}\|_2^2 + \eta\sum_{k}^{K}\left\{\|\mathbf{W}_k * \mathbf{x} - \boldsymbol{\alpha}_k\|_F^2 + J(\boldsymbol{\alpha}_k)\right\}. \tag{12}$$

The iterative process is shown as follows:

$$\begin{cases}
\mathbf{x} = \underset{\mathbf{x}}{\text{argmin}}\|\mathbf{y} - \boldsymbol{\Phi}\mathbf{x}\|_2^2 + \eta\sum_{k}^{K}\left\{\|\mathbf{W}_k * \mathbf{x} - \boldsymbol{\alpha}_k\|_F^2\right\}, \\
\boldsymbol{\alpha}_k = \underset{\boldsymbol{\alpha}}{\text{argmin}}\|\mathbf{W}_k * \mathbf{x} - \boldsymbol{\alpha}_k\|_F^2 + J(\boldsymbol{\alpha}_k).
\end{cases} \tag{13}$$

The first sub-problem can be solved by gradient methods:

$$\mathbf{x}^{(t+1)} = \mathbf{x}^{(t)} - \delta\left(\boldsymbol{\Phi}^\top\left(\boldsymbol{\Phi}\mathbf{x}^{(t)} - \mathbf{y}\right) + \eta\sum_{k}^{K}\left(\mathbf{W}_k^\top\left(\mathbf{W}_k\mathbf{x}^{(t)} - \boldsymbol{\alpha}_k^{(t+1)}\right)\right)\right). \tag{14}$$

Under some assumptions, Equation (14) is simplified as a residual form:

$$x^{(t+1)} \approx \rho \mathbf{x}^{(t)} + \delta \mathbf{x}^{(0)} + \gamma \mathbf{x}^{(t+1/2)}, \tag{15}$$

where $\mathbf{x}^{(t+1/2)} = \sum_k^K \mathbf{W}_k^\top \boldsymbol{\alpha}_k^{(t+1)}$. It is the output of the convolutional layer. Then, the unrolled iterative algorithm is changed to a network. Moreover, the measuring matrix is replaced by a convolutional layer with $m$ channels, $L \times L$ kernel size and $s$ stride. The initial reconstruction is computed by another convolutional layer. All the convolution parameters are learnable.

There are some other works that belong to this category. The authors of [10] proposed Iterative Firm Thresholding Algorithm (IFTA) to solve general inverse problem and set most parameters to be learnable. In [11], the weights of proximal operator is obtained by training. For low-rank tensor factor analysis approach, the authors of [12] used neural networks to substitute the matrix computation.

### 2.3. Neural Networks as Image Projections

The second category is to directly use neural networks (or some deep learning modules) as the $P_{\text{image}}$. In this category, an initial reconstructed image is needed. In some works, the initial reconstruction is contained in the beginning part of neural networks. At first, the reconstruction model only contained one $P_{\text{image}}$ and no $P_{\text{dc}}$, just like a denoising model. Later, more sophisticated networks were proposed and $P_{\text{dc}}$ was included as one layer of the model. When more than one $P_{\text{image}}$ and $P_{\text{dc}}$ occur in the network, it has an unrolling form similar to traditional iterative algorithms such as alternating direction method of multipliers (ADMM), ISTA and denoising approximate message passing (D-AMP). It is worth noting that when $P_{\text{image}}$ is represented by a neural network, the image prior is hidden. In this category, networks can be regarded as substitutes for the original proximal operator in iterative algorithms. Different algorithms lead to different form of networks. Generally speaking, most improvements are about network architecture and loss function design.

The authors of [13] proposed to use a three-layer fully-connected network to reconstruct image from measurements. The input of network is measurements and the output is reconstructed images. Since fully-connected layers are used, the network is trained with $32 \times 32$ patches to reduce parameters. Correspondingly, the measurements are obtained from image patches. Later, a convolutional neural network was applied in the same manner [14]. The first layer of the CNN is still a fully-connected layer to transform the measurement to image space. In this work, patches with size of $33 \times 33$ were used for training. In addition, block-matching and 3D filtering (BM3D) [15] is exploited as post-process to overcome the blocky artifacts when reconstructed image patches are pieced together to form the whole image. In [16], the first fully-connected layer of the CNN is replaced by the transposition of measuring matrix. The CNN proposed by [17] contains not only the reconstruction part but also the measuring part. One convolutional layer with $n_B$ channels, $B \times B$ kernel size and $B$ stride plays the role of measuring. It is followed by a convolutional layer with $B^2$ channels and $1 \times 1$ kernel size which is used to reconstruct image initially. The output is then reshaped to the original image size. In fact, such measuring and initial reconstructing manner is equivalent to block CS. However, it makes it possible that the whole image can be fed into the network.

Residual structure [18] was applied to reconstruct image [19]. The first layer of the network is a fully-connected layer to transform measurements into image space. The following part contains several residual learning blocks. Similar to [14], patches are used for training and BM3D is also exploited to remove blocky artefacts. The training scheme is composed of two steps. Firstly, the fully-connected layer is trained using mean squared error (MSE) loss. Then the whole network is trained in an end-to-end manner. The authors of [20] proposed a similar method, but the measuring matrix is replaced by one fully-connected layer and also trained together with other part of network. In [21], the BM3D module is substituted by another residual convolutional block which is the combination of $11 \times 11, 1 \times 1, 7 \times 7$

convolutions and ReLU functions. Different from [19], in [22] the measuring and initial reconstructing parts are convolutional and deconvolutional layers, respectively, so as to reconstruct the whole image instead of patches.

The choice for loss functions is also explored in some works. Besides popular MSE (L2 norm) loss, adversarial loss [23] is used when training networks [24]. It was proposed firstly to train generative adversarial networks (GAN). The basic network architecture is an analogy to the reconstruction part in [21]. Perceptual loss is exploited in [25] and structural similarity (SSIM) loss is applied in [26]. All these loss functions are used to enhance the quality of reconstructed images.

More works focus on how to design the network architecture to achieve better reconstruction performance. A two-branch network was proposed by [27]. One branch utilizes dense connection structures and the other one consists of residual blocks. Random sampling scheme and fully-connected sampling scheme are all considered. Since it is based on block CS, BM3D is also used to remove blocky artefacts after patch reconstruction. The authors of [28] proposed a pyramid-structured adversarial network. In general CS problem, reconstructed images have a fixed resolution. As long as the number of measurements is insufficient, the reconstruction quality is unsatisfied. The idea of the pyramid network is that the resolution of reconstructed images depends on the number of measurements. A low-resolution image is reconstructed from fewer measurements while high resolution ones are reconstructed from more measurements. Different levels of resolution correspond to different sub-networks. The input of sub-networks is the reconstructed image from last level and measurements.

Scalable sampling rates are considered in [29] and SCSNet was proposed. Measurements are divided into groups and used as reconstructed information for different scales. One group is used to reconstruct the low frequency part of images which corresponds to the basic layer in network. Others are used to reconstruct the high frequency part corresponding to enhanced layers (EL). Measurements are obtained from image patches by a non-overlapping block convolutional layer. After initial reconstruction, a deep reconstruction network is applied to reconstruct the whole image. In this work, the MSE loss function is applied to both initial reconstruction and final reconstruction.

A recent work exploited the idea that reconstructed signals can be decomposed into two orthogonal parts [30]. One is in the null space of the measuring matrix $\mathbf{H}$ and the other is in the pseudo-inverse space. Suppose the measurements satisfy $\mathbf{y}_\varepsilon = \mathbf{H}\mathbf{x} + \boldsymbol{\varepsilon}$. $\mathbf{x}$ is decomposed by $\mathbf{x} = \mathcal{P}_r(\mathbf{x}) + \mathcal{P}_n(\mathbf{x})$ where $\mathcal{P}_r \triangleq \mathbf{H}^\dagger\mathbf{H}$ and $\mathcal{P}_n \triangleq (\mathbf{I}_D - \mathbf{H}^\dagger\mathbf{H})$. Then we can derive that $\mathbf{x} = \mathbf{H}^\dagger\mathbf{y}_\varepsilon + \mathbf{H}^\dagger\boldsymbol{\varepsilon} + \mathcal{P}_n(\mathbf{x})$. The network consists of two parts which is used to reconstruct the two signal components, respectively. The authors of [30] considered two forms of architectures.

Multi-scale structures were utilized in [31]. There are three branches of sub-networks with different convolutional kernel sizes to extract information of different scales. All the sub-networks have residual blocks and non-local layers which are helpful for global information extraction. At the beginning of training, three sub-networks are trained, respectively, and non-local layers are neglected. Finally, the whole network is trained in an end-to-end manner.

The works reviewed above are all about how to design a neural network as the $P_{\text{image}}$, and no $P_{\text{dc}}$ is used. Some works generalize this approach to use neural network many times and combine it with traditional iterative algorithms to form an unrolling architecture. Since the $P_{\text{dc}}$ is contained in iterative algorithms, the whole network is composed of many $P_{\text{image}}$s and $P_{\text{dc}}$s. Usually, the $P_{\text{dc}}$ retains the original form. In other word, in this approach neural networks substitute the original $P_{\text{image}}$s of the iterative process. The role of $P_{\text{image}}$s in unrolling methods is, in essence, the same to those that only use one $P_{\text{image}}$. Therefore, any network design mentioned above is also applicable. Some works train the $P_{\text{image}}$ network beforehand, while others train the unrolled network in an end-to-end manner. As for the concrete form of $P_{\text{dc}}$, gradient computation is used in some works while the proximal operator is used in others.

The authors of [32] proposed to train a projection network to replace proximal operators in the iterative algorithm. The purpose is to solve all the inverse problem, including CS reconstruction. The projection network plays the role of $P_{\text{image}}$ and is trained beforehand. It contains an auto-encoder $\mathcal{P}$ and two discriminators, $\mathcal{D}$ and $\mathcal{D}_\ell$. The input of auto-encoder is a clean image or a perturbed one which is obtained by adding Gaussian noise. $\mathcal{D}$ is used to discriminate the outputs of $\mathcal{P}$ while $\mathcal{D}_\ell$ is for the encodes of $\mathcal{P}$. The loss function has the following form:

$$\min_{\theta_\mathcal{P}} \sum_{\mathbf{x} \in \mathcal{M}, \mathbf{v} \sim f(\mathbf{x})} \lambda_1 \|\mathbf{x} - \mathcal{P}(\mathbf{x})\|^2 + \lambda_2 \|\mathbf{x} - \mathcal{P}(\mathbf{v})\|^2 + \lambda_3 \|\mathbf{v} - \mathcal{P}(\mathbf{v})\|^2$$
$$- \lambda_4 \log(\sigma(\mathcal{D}_\ell \circ \mathcal{E}(\mathbf{v}))) - \lambda_5 \log(\sigma(\mathcal{D} \circ \mathcal{P}(\mathbf{v}))), \tag{16}$$

where $\mathbf{x}$ is a clean image, $\mathbf{v}$ is the perturbed one and $\mathcal{E}$ is the encoder of $\mathcal{P}$. In this work, ADMM algorithm is used and the trained projection network substitutes the first step of iteration. The authors of [33] also proposed to use a neural network to represent the proximal operator in ADMM algorithm. However, they utilized a denoising CNN with residual structures and different noisy level are tested to attain the best performance. Similarly, the proximal operator in proximal gradient method is replaced by a neural network in [34]. The authors of [35] proposed to use a neural network as denoising model in D-AMP algorithm. The modified D-AMP algorithm has the following form:

$$\mathbf{b}^t = \frac{\mathbf{z}^{t-1} \operatorname{div} D_{\hat{\sigma}^{t-1}}\left(\mathbf{x}^{t-1} + \mathbf{A}^H \mathbf{z}^{t-1}\right)}{m}, \tag{17}$$

$$\mathbf{z}^t = \mathbf{y} - \mathbf{A}\mathbf{x}^t + \mathbf{b}^t, \tag{18}$$

$$\hat{\sigma}^t = \frac{\|\mathbf{z}^t\|_2}{\sqrt{m}}, \tag{19}$$

$$\mathbf{x}^{t+1} = D_{\hat{\sigma}^t}\left(\mathbf{x}^t + \mathbf{A}^H \mathbf{z}^t\right). \tag{20}$$

where $D_{\hat{\sigma}^{t-1}}$ is the neural network. The D-AMP algorithm was also applied to block CS reconstruction in [36]. The denoising model, i.e., $P_{\text{image}}$ is a DnCNN [37]. For efficiently sampling, the sampling rate of patches depends on the salient value. Patches with the same value are measured by the same measuring matrix which, specifically, is a convolutional layer. $P_{\text{dc}}$ is computed for patches while $P_{\text{image}}$ is computed for the whole image.

The authors of [38] proposed to treat a neural network as a regularization term. The model is written as:

$$\mathbf{x}_{\text{rec}} = \arg\min_{\mathbf{x}} \underbrace{\|\mathcal{A}(\mathbf{x}) - \mathbf{b}\|_2^2}_{\text{data consistency}} + \lambda \underbrace{\|\mathcal{N}_{\mathbf{w}}(\mathbf{x})\|^2}_{\text{regularization}}. \tag{21}$$

where $\mathcal{N}_{\mathbf{w}}(\mathbf{x}) = (\mathcal{I} - \mathcal{D}_{\mathbf{w}})(\mathbf{x}) = \mathbf{x} - \mathcal{D}_{\mathbf{w}}(\mathbf{x})$ and $\mathcal{D}_{\mathbf{w}}(\mathbf{x})$ represents a neural network. Then the unrolling architecture can be derived as follows:

$$\mathbf{x}_{n+1} = \arg\min_{\mathbf{x}} \|\mathcal{A}(\mathbf{x}) - \mathbf{b}\|_2^2 + \lambda \|\mathbf{x} - \mathbf{z}_n\|^2, \tag{22}$$

$$\mathbf{z}_n = \mathcal{D}_{\mathbf{w}}(\mathbf{x}_n). \tag{23}$$

The first step corresponds to the $P_{\text{dc}}$ to keep data consistency. While the neural network $\mathcal{D}_{\mathbf{w}}$ is the $P_{\text{image}}$. To reduce parameters, weights of the denoiser in all iterations are shared. The training scheme contains two stages. In the first stage, only one iteration is trained. In the second stage, all the iterations with shared weights are trained together. A similar approach was proposed in [39] and the authors hold a viewpoint that residual structure is feasible to represent prior. The authors of [40] proposed an unrolling network based on a primal-dual algorithm where proximal operators are replaced by a three-layer network

with PReLU activation functions. An extra gradient method was unrolled in [41] and Nesterov's accelerated gradient method was utilized.

The authors of [42] proposed a Network-based PGD (NPGD) method to reconstruct images from CS measurements. The $P_{\text{image}}$ in this work is not a denoising model, but a composition of a GAN and its inverse network. Firstly, a trained GAN is used for depicting image prior. The generator is denoted by $G$ and its inverse network $G^{\dagger}$ is trained to project a image signal to the latent space of $G$. Thus, $G \circ G^{\dagger}$ plays the role of $P_{\text{image}}$. The following loss function was proposed to train $G^{\dagger}$,

$$\mathcal{L}(\theta) = \mathbb{E}_{z,\nu}\left[\left\|G\left(G^{\dagger}_{\theta}(G(z) + \nu)\right) - G(z)\right\|^{2}\right] + \mathbb{E}_{z,\nu}\left[\lambda\left\|G^{\dagger}_{\theta}(G(z) + \nu) - z\right\|^{2}\right]. \tag{24}$$

Based on unrolling networks, some works focus on improvements of denoising models. The authors of [43] divided the model into three sub-models which are based on MWCNN [44]. Each one deals with different levels of noise and their average output is used finally. In addition, the input of sub-models is expanded in channels and each channel keeps identical. Usually, the input of the $P_{\text{image}}$ is a corrupted image. However, in [45] image is decomposed into several combinations of high frequency parts and corresponding low frequency ones. Only high frequency ones are input. After denoising, the clean high frequency parts are added to corresponding low frequency one. Finally, the average of different combination is the output of the $P_{\text{image}}$. Frequency decomposition is realized through minimizing an object function consist of a total variation with different coefficients. The coefficients control the frequency decomposition.

### 2.4. Latent Variable Search of Generative Models

The third category is the latent variable search of the generative model. The basic idea is simple. Firstly, a generative model, such as GAN, is trained. Its output represents the image prior manifold. Then, minimize a loss function, which usually corresponds to the data consistency, by searching the latent variable. Generally speaking, the object of CS reconstruction is to find a best image **x**. However, in this category of methods the search of **x** is replaced by the search of latent space variable. Suppose the trained generative model is $G$, latent variable is **z**, and data consistency is represented by $\|\mathbf{y} - \mathbf{A}\mathbf{x}\|_{2}^{2}$. Then the optimization problem is as follows:

$$\min_{\mathbf{x}}\|\mathbf{y} - \mathbf{A}\mathbf{x}\|_{2}^{2}, \quad \text{s.t } \mathbf{x} = G(\mathbf{z}). \tag{25}$$

It can also be rewritten as follows:

$$\min_{\mathbf{z}}\|\mathbf{y} - \mathbf{A}G(\mathbf{z})\|_{2}^{2}. \tag{26}$$

When the solution $\mathbf{z}^{*}$ is obtained, reconstruction result is derived by $G(\mathbf{z}^{*})$. At first look, it is hard to verify that this method also satisfies framework $\mathcal{F}$. Suppose that we use a simple first order gradient method to solve Equation (26), we have the following decomposition for each iteration by the chain rule:

$$\mathbf{z}^{(k+1)} = \mathbf{z}^{(k)} - \eta\frac{\partial\left\|\mathbf{A}G\left(\mathbf{z}^{(k)}\right) - \mathbf{y}\right\|_{2}^{2}}{\partial\mathbf{z}^{(k)}} \tag{27}$$

$$= \mathbf{z}^{(k)} - \eta\mathbf{D}^{(k+1)}\mathbf{r}^{(k+1)}, \tag{28}$$

where $\mathbf{r}^{(k+1)} = \frac{\partial\left\|\mathbf{A}G\left(\mathbf{z}^{(k)}\right) - \mathbf{y}\right\|_{2}^{2}}{\partial G\left(\mathbf{z}^{(k)}\right)}, \mathbf{D}^{(k+1)} = \frac{\partial G\left(\mathbf{z}^{(k)}\right)}{\partial\mathbf{z}^{(k)}}$. In fact, the generative model represents $M_{\text{image}}$, and the composition of $P_{\text{image}}$ and $P_{\text{dc}}$ can be represented by $\mathcal{P}_{\text{image}} \circ \mathcal{P}_{\text{dc}} = \mathcal{V}_{x} \circ \mathcal{P}_{x} \circ \mathcal{U}_{x}$ where $\mathcal{U}_{x} = G^{-1}(\mathbf{x})$, $\mathcal{P}_{x} = \mathbf{z} - \eta\mathbf{D}\mathbf{r}$ and $\mathcal{V}_{x} = G(\mathbf{z})$. $P_{dc}$ is implied by the loss function and is hidden in the derivative computation of **r**. Actually, **r** corresponds to $P_{dc}$.

Figure 3 shows the movement of **x**. After computing **r**, the direction is limited to the latent space by **Dr**. Then through the generative model, the limited direction corresponds to the movement of **x** and $P_{\text{image}}$ is realized. Thus, this category also satisfies framework $\mathcal{F}$.

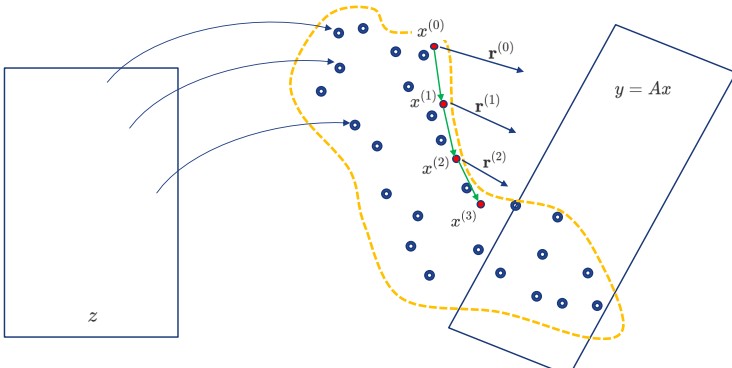

**Figure 3.** An illustration for latent variable search of generative models. The dash line and scatter points represent $M_{image}$. The blue arrows stand for **r** and generative model limits the movement direction of **x** in $M_{image}$

This approach was firstly proposed in [46] and two theorems about the error upper bound are given. An improvement was proposed by the authors of [47]. A sparse item is added to correct the reconstruction. The object function has the following form:

$$\min_{\mathbf{z},\mathbf{v}} \|\mathbf{v}\|_0, \tag{29}$$

$$\text{s.t. } \mathbf{A}(G(\mathbf{z}) + \mathbf{v}) = \mathbf{y}. \tag{30}$$

Then CS problem is solved by minimizing a non-constraint object function as follows using a first-order gradient method:

$$\min_{\mathbf{z},\mathbf{v}} \|\mathbf{v}\|_1 + \lambda \|\mathbf{A}(G(\mathbf{z}) + \mathbf{v}) - \mathbf{y}\|_2^2. \tag{31}$$

Here, zero norm is replaced by L1 norm. Another improvement in [48] is that the latent variable is also optimized when training the GAN model. In other word, a set of latent variable $\{\hat{\mathbf{z}}^{(1)}, \hat{\mathbf{z}}^{(2)}, \ldots, \hat{\mathbf{z}}^{(s)}\}$ is trained to satisfy that $\mathbf{y}^{(i)} = \mathbf{A}G(\hat{\mathbf{z}}^{(i)})$. When training data is insufficient, another discriminator is applied to discriminate measurements besides the usual image discriminator. Auto-encoders and generative models are combined in [49]. Auto-encoders tend to effectively extract low-frequency structure of image while losing details. GANs are good at generating images with fine details but may cause global corruption. Thus, the fitting in measurement in Equation (26) is substituted by encode fitting. In addition, $\|\mathbf{z}\|_2^2$ is added as a regularization term.

Instead of first-order gradient method, ADMM algorithm is also used to solve Equation (26). In [50], suppose there is a regularization of **z** denoted by $H(\mathbf{z})$. Then the object function is $\min_{\mathbf{x},\mathbf{z}} \|\mathbf{y} - \mathbf{\Phi}\mathbf{x}\|_2^2 + \lambda H(\mathbf{z})$, s.t. $\mathbf{x} = G(\mathbf{z})$. The iteration has the following form:

$$\mathbf{x}^{(k+1)} = \left(\mathbf{\Phi}^T\mathbf{\Phi} + \rho\mathbf{I}\right)^{-1}\left(\mathbf{\Phi}^T\mathbf{y} + \rho\left(G\left(\mathbf{z}^{(k)}\right) - \boldsymbol{\mu}^{(k)}\right)\right), \tag{32}$$

$$\mathbf{z}^{(k+1)} = \arg\min_{\mathbf{z}} \lambda H(\mathbf{z}) + \frac{\rho}{2}\left\|\mathbf{x}^{(k+1)} - G(\mathbf{z}) + \boldsymbol{\mu}^{(k)}\right\|_2^2, \tag{33}$$

$$\boldsymbol{\mu}^{(k+1)} = \boldsymbol{\mu}^{(k)} + \mathbf{x}^{(k+1)} - G\left(\mathbf{z}^{(k+1)}\right). \tag{34}$$

To solve the second step, a fully-connected network $G_{\text{proj}}$ was proposed in [50]. It has to be trained using pairs of $(\tilde{\mathbf{x}}, \mathbf{z})$ where $\tilde{\mathbf{x}}$ is a noisy signal represented by $\tilde{\mathbf{x}} = G(\mathbf{z}) + \varepsilon$. Another trick in training the GAN is that each latent variable **z** is split into code-words

$c$ and "random-noise-like" variable $\gamma$, which is inspired from InfoGAN [51]. $c$ is used to control the semantic information and $\gamma$ controls variation. A loss function that maximize the mutual information between $c$ and $G(\mathbf{z})$ is included. The authors of [52] proposed a new training strategy combining meta learning and generative model to accelerate the search of latent variable.

### 2.5. Neural Networks Based Probability Models

The fourth category is to use neural network to represent prior distribution of images and maximize the posterior probability. It is one of Bayesian models and the projection direction of the $P_{\text{image}}$ and the $P_{\text{dc}}$ are related to $\nabla_x \log p(x)$ and $\nabla_x \log p(y|x)$. Thus, this category satisfies Framework $\mathcal{F}$.

RIDE model was proposed in [53]. It combines a LSTM [54] model and Mixture of Conditional Gaussian Scale Mixtures as image prior distribution. Then the gradient method is used to solve the posterior distribution. Later, in [55] a PixelCNN [56] was applied to represent image prior. Its model has the following form:

$$p(\mathbf{x}) = p(x_1, x_2, \ldots, x_{n^2}) = \prod_{i=1}^{n^2} p(x_i | \mathbf{x}_{<i})). \tag{35}$$

### 2.6. Unsupervised Methods

Last category is unsupervised method. When there is no real image dataset, it is hard to depict image prior. In [57], deep image prior (DIP) method was proposed. It was used to solve some inverse problem not including CS. Later, DIP was applied to reconstruct image from compressed measurements. Most of current unsupervised methods are based on it. The basic idea is to use an untrained generative model and minimize the loss function of data consistency over network parameters with fixed input $\mathbf{z}$. DIP method is similar to the category discussed in Section 2.4. However, $M_{\text{image}}$ is represented by an untrained network itself instead of a trained generative model. In other words, image prior is depicted by the output of generative model with fixed latent variable and learnable network parameters. Searching in parameter space is analogue to searching in latent space. Thus, similar analysis of Section 2.4 can be used to verify that this category also satisfies framework $\mathcal{F}$.

The authors of [58] proposed to applied DIP method to solve CS reconstruction. A regularization term is added to loss function which has the following form:

$$\arg\min_{w} \|\mathbf{y} - \mathbf{A}G(\mathbf{z}; w)\|^2 + R(G(\mathbf{z}; w), w; \lambda_T, \lambda_L), \tag{36}$$

where $R(G(\mathbf{z}; w), w; \lambda_T, \lambda_L) = \lambda_T TV(G(\mathbf{z}; w)) + \lambda_L (w - \mu)^T \Sigma^{-1}(w - \mu)$. $\mu$ and $\Sigma$ are the mean and covariance matrix of network parameters estimated by a few data. Total variation regularization was used in [59] to help reconstruction. In [60], semi-supervised learning was discussed. In Section 2.4, training a generative usually need a great deal of data. The authors of [60] proposed a strategy to make a trade-off. In pre-train stage, network parameters and latent variables are trained simultaneously with a combination of image L2 loss and kernel loss. The latter has the following form:

$$\min_{\boldsymbol{\theta}, \mathbf{z}_1, \ldots, \mathbf{z}_S} \frac{1}{\binom{S}{2}} \sum_{i \neq i'} k(G(\mathbf{z}_i; \boldsymbol{\theta}), G(\mathbf{z}_i; \boldsymbol{\theta})) + \frac{1}{\binom{S}{2}} \sum_{j \neq j'} k\left(\mathbf{x}_j, \mathbf{x}_{j'}\right) - \frac{2}{\binom{S}{2}} k(G(\mathbf{z}_i; \boldsymbol{\theta}), \mathbf{x}_j), \tag{37}$$

where $\mathbf{z}_i$ is the latent variable, $\boldsymbol{\theta}$ is network parameters and $S$ is the number of training sample. In the reconstruction stage, latent variable is first optimized and then together with network parameters.

### 2.7. Discussion

In this section, we further explained different categories of deep learning methods by our framework, especially latent variable search of generative models. We can observe a trend from simple networks to complex and bigger ones. Among these methods, cascaded networks, which serve as image projections perform best. While generative model or probability model based methods are less comparable due to the unsatisfied performance of generative models and probability model. However, they still have potential to be improved in the future as more powerful generative models and probability models are proposed.

## 3. Deep Learning Methods for Computed Tomography

### 3.1. Overview

We mainly discuss sparse-view or limited angles CT reconstruction in this section. All the works reviews here belong to the five categories of Section 2. Therefore, they must satisfy framework $\mathcal{F}$. The emphasis is to illustrate how they design $P_{\text{image}}$ and $P_{\text{dc}}$. It is worthy of mention that the initial reconstruction is obtained by the FBP algorithm.

### 3.2. Model-Based Methods with Learnable Parts

There are few works belonging to this category. The authors of [61] proposed to applied variational network to low-dose CT reconstruction. Fields of experts are used as regularization term of variational model shown as follows:

$$R_c(\mathbf{u}) = \langle 1, \phi_c(K_c\mathbf{u}; W_c) \rangle \tag{38}$$

where $\mathbf{u}$ is a CT image, $\phi_c$ is a linear interpolation and $K_c$ is a convolution. $K_c$ and $W_c$ are learned by training. The network architecture corresponds to unrolling the first-order gradient method:

$$u_t = u_{t-1} - K_c^\top \phi_c'(K_c u_{t-1}; W_c) - \lambda_c A^\top (A u_{t-1} - d). \tag{39}$$

JSR-net [62] was proposed to solve limited angle CT reconstruction. It unrolled the ADMM algorithm for JSR-model. The computation of two inverse matrices and the thresholding function in JSR-model are replaced by neural networks. The former is substituted by a three-level DenseNet with LM-ResNet structure and the latter by a three-layer convolutional network. The object function of JSR model has the following form,

$$\min_{\mathbf{u},\mathbf{f}} F(\mathbf{u}, \mathbf{f}, \mathbf{Y}) + \|\lambda_1 \mathbf{W}_1 \mathbf{u}\|_{1,2} + \|\lambda_2 \mathbf{W}_2 \mathbf{f}\|_{1,2}, \tag{40}$$

where

$$F(\mathbf{u}, \mathbf{f}, \mathbf{Y}) = \frac{1}{2}\|R_{\Gamma^c}(\mathbf{f} - \mathbf{Y})\|^2 + \frac{\alpha}{2}\|R_\Gamma(\mathcal{P}\mathbf{u} - \mathbf{f})\|^2 + \frac{\gamma}{2}\|R_{\Gamma^c}(\mathcal{P}\mathbf{u} - \mathbf{Y})\|^2, \tag{41}$$

and $\Gamma^c$ represents sampling angles.

### 3.3. Neural Networks as Image Projections

Most works of CT reconstruction using deep learning methods belong to this category. The authors of [63] applied a fully-connected network to refine the middle result of traditional iterative algorithms. A three-layer CNN was used to low-dose CT reconstruction task in [64,65]. The input of network is initial reconstruction of the FBP algorithm. The authors of [66] proposed to applied a U-Net for reconstruction. Residual structures were considered in [67]. Later, the authors of [68] proposed to add bypass connections and utilize the Haar wavelet as down-sampling and up-sampling to improve the reconstruction.

Almost all of methods employ L2 or L1 norm loss of image. Some works also apply other type of loss functions. Perceptual loss was used in [69]. Adversarial loss function was exploited in [70]. A discriminator was used to help to refine the details of reconstruction by adversarial training.

Some works change the object of $P_{\text{image}}$. It means that there are explicit $U_x$ and $V_x$. In [71], input of the U-Net is the result of wavelet decomposition of the initial reconstruction image which purpose is to utilize multi-scale information. In other word, $U_x$ and $V_x$ is related to wavelet decomposition and synthesis. Using sinogram measurements as network inputs was proposed by [72]. For different angles of view, corresponding sinogram was expand to a image by back projection and these images were stacked to form a tensor. Then it was used as input of a 15-layer CNN. In this work, $U_x$ is the process of sinogram and $V_x$ is merge into $P_x$. In [73], interpolated sinogram was used as the input of a U-Net and the output is the accurate sinogram. When output of network is obtained, FBP algorithm is applied to compute the final reconstruction. In this method, $U_x$ is the transform from initial image to sinogram space, $V_x$ is executed by the FBP algorithm and $P_x$ is the U-Net. Thus, the projection operator is executed in sinogram space. This is an example illustrating a difference between medical application and general CS problem. In fact, sinogram is the measurement $\mathbf{y}$ and there exists a transform and its inverse between measurement space and image space. We can also regard the method in [73] as a projection model in measurement space. Similar to [73,74] used a U-Net to reconstruct under-sampled sinogram. Besides, a discriminator and adversarial training was exploited in this work. The input of the discriminator is the sinogram with limited angles and full-size output of generative.

All the works mentioned above only consist of one $P_{\text{image}}$ and no $P_{\text{dc}}$. Similar methods can be seen in [75–81]. Next, deep learning methods with more than one $P_{\text{image}}$ and $P_{\text{dc}}$ will be reviewed.

The authors of [82] considered a regularization term of Fields of Experts and used a simple first-order gradient method to solve the object function. It has the following form:

$$\mathbf{x}^{t+1} = \mathbf{x}^t - \left( \lambda^t \mathbf{A}^T (\mathbf{A}\mathbf{x}^t - \mathbf{y}) + \sum_{k=1}^{K} (G_k^t)^T \gamma_k^t (G_k^t \mathbf{x}^t) \right), \tag{42}$$

where $\sum_{k=1}^{K} (G_k^t)^T \gamma_k^t (G_k^t \mathbf{x}^t)$ is related to Fields of Experts. This term was replaced by a three-layer CNN which plays the role of the $P_{\text{image}}$. Since Equation (42) is in an iterative form, the network contains many $P_{\text{image}}$s and $P_{\text{dc}}$s.

The authors of [83] proposed to unroll the ADMM algorithm and added a regularization term about sinogram to original object function. Thus, there are two types of $P_{\text{image}}$s. The object function is shown as follows:

$$\min_{\mathbf{x},\mathbf{y}} \frac{1}{2} \|\mathbf{y} - \hat{\mathbf{y}}\|_{\Sigma_y^{-1}}^2 + \frac{1}{2} \|\mathbf{A}x - y\|_{\Sigma_x^{-1}}^2 + \lambda R_y(\mathbf{y}) + \gamma R_x(\mathbf{x}). \tag{43}$$

Though there is an explicit transform relationship between sinogram $\mathbf{y}$ and image $\mathbf{x}$, in the optimization task they are split to exploit $\lambda R_y(y)$. The sinogram regularization is $R_y = \frac{1}{2} \sum_j \sum_{m \in N_j} \omega_{jm} (y_j - y_m)^2$. When the iteration is unrolled into a network, a ResNet was applied to deal with $\gamma R_x(\mathbf{x})$. Besides, L2 norm loss with weights (indicated by $\Sigma_y$ and $\Sigma_x$) was used for $\mathbf{y}$ and $\mathbf{x}$. In [84], unrolling the ADMM network was also used. However, proximal operator of the regularization term was substituted by a U-Net. In [85], a more complex object function is solved by unrolling the ADMM network. The ADMM iteration contains four proximal operators and they were all replaced by three-layer CNNs. That is to say, both $P_{\text{image}}$s and $P_{\text{dc}}$s are represented by neural networks.

The authors of [86] used a denoising auto-encoder with soft-thresholding function as $P_{\text{image}}$ and solved the $P_{\text{dc}}$ by FISTA. In each stage, a cleaner image $\mathbf{z}$ is obtained by the denoising model and FISTA is used to keep data fidelity of $\mathbf{z}$. Because FISTA is not unrolled into a network, the parameters in the $P_{\text{image}}$ cannot be trained in an end-to-end manner. A stage-wise training scheme was proposed to solve the problem.

In [87], a proximal forward backward splitting algorithm was unrolled into a network to reconstruct CT image. It is similar to ISTA network and the proximal operator is replaced

by a CNN. However, instead of last iteration result, all iteration results before are used as the input of CNN in next iteration. In addition, the pseudo-inverse of measuring matrix rather than transposition is used to compute the $P_{dc}$.

Scale invariant property was exploited in [88]. It is combined with the unrolling network. Specifically, the granularity in each iteration becomes finer and in last iteration the original full measuring matrix is used. The iteration has the following form:

$$\begin{cases} f_i = \Lambda_{\theta_i}(\tilde{f}_i, \nabla \mathcal{D}_i(\tilde{f}_i; g)), \\ \tilde{f}_{i+1} = \tau_{i+1}(f_i), \end{cases} \tag{44}$$

where $\nabla \mathcal{D}_i(f_i; g) := \mathcal{A}_i^*(\mathcal{A}_i(f_i) - \pi_i(g))$, $\Lambda_{\theta_i}$ corresponds to the $P_{image}$ and $\tau_{i+1}$ is upsampling operator. The multi-scale idea is similar to multi-level structure in U-Net. Thus, the unrolling form is represented by a U-Net.

In [89], both CNNs and traditional algorithms are used to reconstruct CT image. The methods are used alternatively to improve reconstruction. The recurrent scheme means that there are two types of $P_{image}$ (CNNs and the regularization term in iterative algorithms) and one type of $P_{dc}$. FBPConvNet is chosen as the neural network structure and PWLS-EP or PWLS-ULTRA is the choice for the iterative algorithm. Similar to [86], the training scheme is a stage-wise process.

Other deep learning method in unrolling form can be seen in [90,91] and etc.

### 3.4. Discussion

Most works on CT reconstruction are very close to solve a denoising problem. We found that there are few works that focus on latent variable search of generative models or probability model due to the complexity of the measurement matrix. How to design effective algorithm to combine generative models or probability models and CT reconstruction is an interesting direction in this area.

### 4. Deep Learning Methods for Magnetic Resonance Imaging

#### 4.1. Overview

In this section, we focus on under-sampled MRI reconstruction which is an important application of CS reconstruction. Some properties of MRI reconstruction distinguish it from other CS problem. Firstly, the image is in the field of complex number. In MRI reconstruction, the measurement is called k-space coefficient which is, in fact, the result of the Fourier transform of image. Thus, the measurement and image are represented by complex number. The magnitude of image is used to show the image. For traditional iterative methods, the operations of real number are easy generalized to complex number. However, how to deal with complex number for neural network is a problem since it is based on tensor operations. For most works using deep learning methods, complex numbers are represented by two-channels tensors, i.e., any $x \in \mathbb{C}^n$ is regarded as in $\mathbb{R}^{2n}$. This treatment is equivalent to regard complex number images as two-channel real number images and all the computation is based on real numbers. Another kind of method is to simultaneously keep the complex number operation and use two-channel representation [92]. Secondly, there is a special imaging method called parallel imaging which makes the linear model more complex. In parallel imaging, several coils are utilized and each one corresponds to a k-space measurement, respectively. If every coil is under-sampled, reconstruction images will be still of high quality while reducing scan time. However, the acceleration of this method is limited. Combining with CS reconstruction can further accelerate the scan. In addition, for each coil there is a sensitive matrix differing in every scan and relating to the k-space measurements. The forward model has the following form:

$$\mathbf{y}_i = \mathbf{A}\mathbf{S}_i\mathbf{x} = \mathbf{M}F\mathbf{S}_i\mathbf{x}, i = 1, 2, \dots, c, \tag{45}$$

where $\mathbf{A}$ represents the Fourier transform $F$ with under-sampled mask $\mathbf{M}$, $\mathbf{S}_i$ is sensitive matrix for the $i$th coil and $c$ is the number of coils. Since sensitive matrices are not fixed

parameters, they have to be estimated when reconstruction. A common approach is to estimate sensitive matrices beforehand using SENSE [93] or other algorithms and then regard $\mathbf{AS}_i$ as a fixed measuring matrix. Therefore, each coil has its own data consistency. This is the main difference between single-coil imaging and parallel imaging.

Besides, it is worthy of mention that the $P_{dc}$ in MRI has a very popular form as follows:

$$\hat{y}_j = \begin{cases} F(\mathcal{N}(\mathbf{x}))_j, & \text{if } j \notin \Omega, \\ \frac{F(\mathcal{N}(\mathbf{x}))_j + \lambda y_j}{1+\lambda}, & \text{if } j \in \Omega. \end{cases} \tag{46}$$

where $\Omega$ is the sampled position, $\mathcal{N}(\mathbf{x})$ is the current reconstructed result and $F$ is the Fourier transform. $\lambda$ is the weight to control the extent of data consistency. When $\lambda = \infty$, original sampled k-space coefficients will be retained.

In later sections, deep learning methods will be also divided into several categories and the criterion is the same to Section 2. Each categories will be further divided into non-parallel imaging and parallel imaging sub-categories if necessary.

*4.2. Model-Based Methods with Learnable Parts*

4.2.1. Non-Parallel Imaging

In [94], the original objection function is as follows:

$$\hat{\mathbf{x}} = \arg\min_{\mathbf{x}} \left\{ \frac{1}{2}\|\mathbf{Ax} - \mathbf{y}\|_2^2 + \sum_{l=1}^{L} \lambda_l g(D_l \mathbf{x}) \right\}. \tag{47}$$

Based on ADMM algorithm, the linear transform $D_l$ in regularization terms was replaced by learnable convolutions and the shrinkage function in iterations was substituted by a learnable piece-wise linear function. step lengths is also learnable parameters. Later, the authors proposed another form of the ADMM network in [95]. Similar to [94,96] also unrolled a ADMM network. Because the noise model is supposed to be symmetric $\alpha$-stable, therefore L1 norm loss is adopted. In practical terms, a smoothing term is used to replace L1 norm. IFR-CS model was proposed in [97] which network architecture is based on [7]. Besides data consistency and proximal operator of the regularization term, a refine step was added in the iterations.

4.2.2. Parallel Imaging

The authors of [98] proposed a variational network which is based on the Fields of Experts model. The regularization term in this model $\mathcal{R}(\mathbf{u})$ is $\sum_{i=1}^{N_k} \langle \Phi_i(\mathbf{K}_i \mathbf{u}), \mathbf{1} \rangle$. The first-order gradient method is used to solve the original object function which has the following form:

$$\mathbf{u}^{t+1} = \mathbf{u}^t - \sum_{i=1}^{N_k} (\mathbf{K}_i^t)^\top \Phi_i^{t\prime}(\mathbf{K}_i^t \mathbf{u}^t) - \lambda^t \mathbf{A}^*(\mathbf{Au}^t - \mathbf{y}), \quad 0 \le t \le T - 1. \tag{48}$$

All the parameters are learnable including $\mathbf{K}, \Phi_i$ and $\lambda$. $\Phi_i^{t\prime}$ is represented by Gaussian radius basis functions. As for sensitive matrices, they are estimated by ESPIRiT [99] algorithm beforehand. The authors of [100] considered parallel imaging and their method is similar to [94]. There are also some works such as [101–103] that can be put under this category.

*4.3. Neural Networks as Image Projections*

4.3.1. Non-Parallel Imaging

The authors of [104] may be the first to apply deep learning in MRI reconstruction. Their method is to train a three-layer CNN beforehand and use the network to reconstruct image. Three ways were proposed in [104]. The first one is to minimize the following object function:

$$\left\| C\left(\mathbf{A}^{H}\mathbf{y}; \hat{\Theta}\right) - \mathbf{x} \right\|_{2}^{2} + \lambda \|\mathbf{y} - \mathbf{A}\mathbf{x}\|_{2}^{2} \tag{49}$$

where $C$ is the trained network. $C$ plays the role of the $P_{\text{image}}$ and the minimization of Equation (49) corresponds to $P_{\text{dc}}$. The second one is to add an extra regularization to Equation (49). In the third way, the output of $C(\mathbf{A}^{H}\mathbf{y}; \hat{\Theta})$ is used as the initial value for a traditional CS reconstruction algorithm.

The authors of [105] proposed to reconstruct the magnitude and phase of image, respectively. The network architecture is a U-Net with global residual learning. Since the value of phase in noisy district is random and meaningless, magnitude network is trained first to ascertain ROI and phase network is trained only in ROI.

The authors of [106] proposed to use conditional GAN to reconstruct MRI whose generative network is U-Net. This method is, in essence, adding adversarial loss function to train the $P_{\text{image}}$. Besides MSE loss of images and adversarial loss, perceptual loss is also used. The authors of [107] applied SSIM loss to dynamic MRI reconstruction. Later, in [108], MSE loss of k-space is added. The authors of [109] proposed to utilize dense connection structure in the bottleneck part of U-Net. The authors of [110] proposed to use adversarial loss function in LSGAN [111]. In addition, the weighted average of L1 norm and L2 norm loss function is utilized. To make training stable, the weight of adversarial loss is set to zero at the beginning of training. Some works modified the structure of U-Net to improve the reconstruction. In [112], convolutions of different sizes were exploited to extract multi-scale information. The features extracted from different convolutions are fused to be the input of the next layer. In [113], dilated convolutions were utilized and residual learning structure was added in the bottleneck of the U-Net. In [114] two U-Nets with residual structures were connected sequentially as the generative network.

Since $P_{\text{dc}}$ is easy to implement, many works whose method contains only one $P_{\text{image}}$ also add one $P_{\text{dc}}$ to correct the reconstruction and keep data consistency. For example, ref. [115] proposed to correct k-space coefficients after using a U-Net to reconstruct images. Some works attempt to reconstruct both image and k-space coefficients. The authors of [116] proposed to use a residual U-Net to reconstruct k-space coefficients and another U-Net for images. These two networks are connected by the inverse of Fourier transform. Reconstructing k-space coefficients is similar to reconstruct sinogram of CT, which is discussed in Section 3. Thus, it corresponds to a $P_{\text{image}}$ which is defined in k-space. The authors of [117] proposed to employ four networks for reconstruction which is named by KIKI-net. Two is for images and two for k-space coefficients. The order is k-space, image, k-space and image (KIKI). For each image network, $P_{\text{dc}}$ is added to guarantee data consistency and connect adjacent networks.

There are also many works utilize more than one $P_{\text{image}}$ and $P_{\text{dc}}$ which form an unrolling network. Some works called it cascade structure because it is not necessary to be derived from an iterative algorithm. However, they are similar in essence because of the alternate order between $P_{\text{image}}$s and $P_{\text{dc}}$s. The authors of [118] proposed a network architecture where CNNs and data consistency are connected alternately. Since the reconstruction object is dynamic MRI, multi-frame images are trained simultaneously and 3D convolution is used. In [119], the output of each CNN in the cascade networks are concatenated at last and a convolutional layer is used to obtain the final reconstruction. The authors of [120] combined neural networks and traditional ADMM algorithm. The training and inference process are both in an ADMM iteration form. The authors of [121] proposed to unroll the Chambolle–Pock algorithm and $P_{\text{dc}}$ is also replaced by a four-layer CNN.

In addition, some works made innovations in network design. The authors of [122] proposed to use dilated convolutions and share parameters in the cascade networks. Dense connections were added to an unrolling network in [123]. The authors of [124] proposed to use a cascade network for k-space reconstruction followed by a network for image.

Recent works considered other types of loss function. The authors of [125] considered to use adversarial loss function and proposed a trick to balance different loss functions.

In [126] perceptual loss was used. In addition, attention layers were applied to U-Net as the $P_{\text{image}}$. In [127], three cascade networks are connected sequentially and their output is concatenated to the last convolutional layer. In each cascade network, convolutions with different strides are used to utilized different scale information. Each network is a RNN which is equivalent to an unrolling form network. In [128], multi-contrast MRI reconstruction was considered and a convolution-shared network was proposed.

### 4.3.2. Parallel Imaging

The authors of [129] utilized a U-Net to reconstruct parallel imaging. WGAN [130] was exploited in [131] and three sequentially connected U-Nets were used as the generator. In training stage, MSE loss of image and k-space, adversarial loss and perceptual loss are applied. The authors of [132] considered 3d MRI reconstruction. The proposed network contains two parts, MS-net for feature extraction and RC-net for reconstruction.

As for methods of unrolling form, most works applied similar network architectures in Section 4.3.1 and the main difference is on the $P_{\text{dc}}$ since each coil has its own data consistency equation. The authors of [133] unrolled a proximal gradient algorithm to a network and applied it to 3d MRI reconstruction. CNNs are used to replace proximal operators. A U-Net was used in [134] to substitute the proximal operator in the ADMM algorithm. In [135], CNNs and data consistency layers are connected alternately and two different process of multi-coil were considered. The authors of [136] proposed to unrolled a first-order gradient method and the regularization term in object function is related to a neural network. The authors of [137] applied the method in [43] to parallel MR imaging. The authors of [138] proposed to utilized variable splitting algorithm. The object function has the following form:

$$\min_{\mathbf{m},\mathbf{u},\mathbf{x}_i} \frac{\lambda}{2} \sum_{i=1}^{n_c} \|\mathcal{D}\mathcal{F}\mathbf{x}_i - \mathbf{y}_i\|_2^2 + \mathcal{R}(\mathbf{u}) + \frac{\alpha}{2} \sum_{i=1}^{n_c} \|\mathbf{x}_i - \mathbf{S}_i\mathbf{m}\|_2^2 + \frac{\beta}{2}\|\mathbf{u} - \mathbf{m}\|_2^2, \tag{50}$$

where $\mathcal{D}$ is the sampling matrix and $\mathbf{m}$ is the reconstructed image. $\mathbf{x}_i$ represents the the result of image multiplying by the sensitive matrix of the $i$th coil. A denoiser network is used to replace the computation of $\arg\min_{\mathbf{u}} \frac{\beta}{2}\left\|\mathbf{u} - \mathbf{m}^k\right\|_2^2 + \mathcal{R}(\mathbf{u})$. In [139] complex number operation was combined with neural networks. The authors of [140] proposed to jointly estimate images and sensitive matrices in a unrolling network. The original object function is as follows:

$$\frac{1}{2}\sum_{l}\|M\mathcal{F}V_l - y_l\|_2^2 + \frac{\rho}{2}\sum_{l}\|S_l \odot U - V_l\|_2^2 + \beta\sum_{l}R(S_l) + \lambda P(U), \tag{51}$$

where $U$ represents the image and $S_l$ is the sensitive matrix. The proximal operator of $R(S_l)$ and $P(U)$ of corresponding iteration are substituted by two sub-blocks and each sub-block contains several sub-networks.

In order to guarantee the convergence of the unrolling network, ref. [141] proposed to use a judgement condition to decide whether to receive the result of the neural network $P_{\text{image}}$.

Besides the reviewed works above, deep learning methods consisting of single $P_{\text{image}}$ can be seen in [92,142–157], etc. Other unrolling form deep learning methods can be seen in [158–170], etc.

### 4.4. Latent Variable Search of Generative Models

A recent work [171] belongs to this category, the purpose of which is to reconstruct parallel imaging of MRI. First, a GAN is trained to generate MRI images. When reconstructing images, besides latent variable, parameters of generative network is also optimized. The reconstruction process contains two stage. In the first stage, the following optimization problem is solved:

$$\min_{\mathbf{z} \in \mathbb{R}^d} \frac{1}{2} \|\mathbf{A} G_\theta(\mathbf{z}) - \mathbf{y}\|^2, \quad \text{s.t. } \|\mathbf{z}\| \leq \sqrt{d}. \tag{52}$$

Then, in the second stage, latent variable and parameters are both optimized:

$$\min_{(\mathbf{z}, \theta) \in \mathbb{R}^d \times \mathbb{R}^l} \frac{1}{2} \|\mathbf{A} G_\theta(\mathbf{z}) - \mathbf{y}\|^2, \quad \text{s.t. } \|\mathbf{z}\| \leq \sqrt{d}. \tag{53}$$

In addition, the sensitive matrices are estimated by the ESPIRiT algorithm.

### 4.5. Neural Networks Based Probability Models

The authors of [172] proposed to estimate the prior distribution by a trained VAE [173] and use it to optimize the Bayesian model through projection onto convex sets (POCS) algorithm. PixelCNN++ [174] was used in [175] to represent the prior distribution and a gradient-projection algorithm was applied to solved the Bayesian model.

### 4.6. Unsupervised Methods

In this category, most related works are based on the DIP method which has been reviewed in Section 2.6. The authors of [176] used DIP directly and made no modification. The authors of [177] applied it to dynamic MRI reconstruction and exploited linear interpolation to obtain inputs of the network for continuous multi-frame images. Meanwhile, the authors of [178] used measurements and zero-fill reconstruction as labels to train a network. The loss function is similar to the one in DIP method and has the following form:

$$\mathcal{L}(\mathbf{y}, \hat{\mathbf{y}}) = \alpha \|\mathbf{y} - \hat{\mathbf{y}}\|_1 + \beta \|\mathbf{\Phi}\mathbf{y} - \mathbf{S} \odot \mathbf{\Phi}(\hat{\mathbf{x}})\|_1 + \gamma \|\mathcal{I}_\theta(\mathbf{y}) - \mathcal{I}_\theta(\hat{\mathbf{y}})\|_1. \tag{54}$$

Here, $\mathbf{y}$ is not measurement but zero-fill reconstruction from under-sampled k-space coefficients and $\hat{\mathbf{y}}$ is the reconstructed image. They are used as the input of the network $\mathcal{I}_\theta$. $\hat{\mathbf{x}}$ is the output of $\mathcal{I}_\theta$; $\mathbf{S}$ is the sampling matrix and $\mathbf{\Phi}$ is Fourier transform. Besides the DIP method, in [179] a novel loss function was designed to implement an unsupervised training scheme. The under-sampled k-space index is divided into two groups which is denoted by $\Omega = \Theta \cup \Lambda$. Correspondingly, measurement $y$ and measuring matrix can also be divided into $(y_\Theta, E_\Theta)$ and $(y_\Lambda, E_\Lambda)$. $(y_\Lambda, E_\Lambda)$ is used as labels and $(y_\Theta, E_\Theta)$ as training inputs. Then the loss function has the following form:

$$\frac{1}{N} \sum_{i=1}^{N} \mathcal{L}\left(\mathbf{y}_\Lambda^i, \mathbf{E}_\Lambda^i \left(f\left(\mathbf{y}_\Theta^i, \mathbf{E}_\Theta^i; \boldsymbol{\theta}\right)\right)\right), \tag{55}$$

where $f$ represents the reconstruction algorithm. Though the method of [179] belongs to unsupervised learning, $f$ in this work is an unrolling form network which similar to the ones reviewed in Section 4.3.

### 4.7. Discussion

Different from CT reconstruction, the measurement matrix in MRI reconstruction is very simple (Fourier transform) and computable. Thus, it is more easy to propose various categories of methods for reconstruction similar to CS reconstruction. We observed that reviewed works cover all the categories mentioned in Section 2 and many unrolling methods were proposed. The trend of deep learning method is similar to CS reconstruction. However, the peculiarity of kspace data and parallel imaging distinguish MRI reconstruction from other medical image reconstruction. The sensitive matrics in parallel imaging also present a challenge for researchers.

## 5. Deep Learning Methods for Positron-Emission Tomography

### 5.1. Overview

Positron-emission Tomography is another common medical imaging tool which utilizes radioactive material. It needs some detectors to receive photons emitted by radioactive element. To reduce the risk, the reconstruction from low-dose PET is desired. Different to CT, the number of detectors is not decreased in most low-dose PET. Thus, in low-dose PET reconstruction, the forward model somehow cannot be deemed as a compressed sensing problem. However, many works that focus on the reconstruction of PET employ deep learning methods similar to CS reconstruction. Those methods can also be classified as some categories discussed in Section 2. Therefore, in this section, we still review some related works of low-dose PET reconstruction.

### 5.2. Neural Networks as Image Projections

In [180], the initial reconstruction results are obtained by a traditional method with different weights. Then the patches of those results are fed into a fully-connected network to produce better reconstruction. The authors of [181] proposed to use a U-Net to transform low-dose PET images to the ones of high quality. The global residual learning structure is utilized and L1 norm is used as the loss function. In addition, several adjacent slices are the input of network. The perceptual loss was exploited in [182]. The network is trained by simulated data at first and then refined by real data. The authors of [183] proposed to use a conditional WGAN to perform 3d reconstruction. The backbone of the generative network is a U-Net and the input is a 3d low-dose PET image. At the beginning and final part of the generative network, 3d convolutions are used while in the middle part 2d convolutions are applied. The training scheme includes two stages. In the first stage, MSE and SSIM loss are used to train the generative network, and in the second stage, adversarial loss and perceptual loss are added to train the model. Similar to [183], in [184] a conditional GAN was proposed. The generative network is a 3d U-Net and its input is patches of low-dose PET images. Besides, a multi-GAN refinement treatment was proposed for better performance. The output of former GAN is used as the input of the next one and each GAN is trained one by one. Some works attempted to exploit other modality information to help PET reconstruction. In [185,186], MR images are fed into network as extra input. For PET, the measurement is also called sinogram since it is similar to CT. In some works it is considered to be the input of networks rather than the initial reconstruction. The authors of [187] used sinogram as the input of a conditional GAN whose generative network is still a U-Net. The authors of [188] used it as the input of a CNN and preprocessed the sinogram before feeding it to the network to reduce the effect of random noise.

Some works are proposed to utilize multiple networks or unroll an iterative algorithm. The authors of [189] proposed a Learned Primal-Dual method which contains two types of U-Nets: one for images and the other for sinogram. These U-Nets are ordered alternatively and connected to each other through measuring matrix and its transposition. The authors of [190] proposed to combine a denoising network and an iterative algorithm. A denoising model DnCNN is trained in advance and added to the logarithmic likelihood function as a regularization. The object function has the following form:

$$\sum_{i=1}^{N_m} [Ax]_i + r_i - y_i \log([Ax]_i + r_i) + \frac{\beta}{2} \|x - q \odot f_w(x) - b\|_2^2. \tag{56}$$

Equation (56) is solved by ADMM after variable splitting. THhe authors of [191] proposed MAPEM-Net which unrolls the ADMM algorithm. A U-Net is used to replace the proximal operator of regularization term. The authors of [192] proposed to use a trained conditional GAN as a constraint for the logarithmic likelihood function. The optimization problem is as follows:

$$\max_{x,\alpha} \{\eta L(\mathbf{y}|\mathbf{P}\boldsymbol{\alpha} + \mathbf{s} + \mathbf{r})) + L(\mathbf{y}|\mathbf{Px} + \mathbf{s} + \mathbf{r}))\}, \quad \text{s.t.,} \ \mathbf{x} = \mathbf{f}(\boldsymbol{\alpha}), \tag{57}$$

where $\mathbf{f}$ represents the generative network and $\boldsymbol{\alpha}$ is five slices of low-dose images. ADMM algorithm is applied to solve it.

### 5.3. Latent Variable Search of Generative Models

The method proposed by [193] can be regarded to belong this category. However, there is something different. In [193], the generative model is a denoising U-Net instead of a GAN or VAE. After pre-training the network, it is used as a constraint for logarithmic likelihood function. The optimization problem can be written as follows:

$$\max_{\mathbf{x}} \quad L(\mathbf{y}|\mathbf{x}), \tag{58}$$

$$\text{s.t.} \quad \mathbf{x} = f(\boldsymbol{\alpha}), \tag{59}$$

where $L$ represents the likelihood function, $f$ is the network and $\boldsymbol{\alpha}$ is the input. It is solved by ADMM algorithm.

### 5.4. Unsupervised Methods

In this part, most works also applied DIP method to PET reconstruction. Similar to [193], in [194], a logarithmic likelihood function with a constraint that $\mathbf{x}$ is the output of a network is the object function for optimization. However, the network is untrained and the input is fixed. Then it turns to be a DIP-like problem. L-BFGS algorithm is used to solve it. The authors of [195] applied an almost same framework for PET reconstruction except that the input of network is replace by a related CT or MR image. The authors of [196] proposed to combine DIP and non-negative matrix factorization to reconstruct dynamic PET images. DIP is used for image representation and non-negative matrix factorization for controlling temporal sparsity. The object function has the following form:

$$\underset{\Theta, \mathbf{B}}{\text{minimize}} \mathcal{L} := D_{\mathrm{KL}}\left(\mathbf{Y}\|\mathbf{PAB}^T\right) + \alpha \left\|\mathbf{A}^T\right\|_{p,2}^2 + \beta \|\mathbf{B}\|_{\mathrm{QV}}^2, \tag{60}$$

$$\text{s.t.} \quad \mathbf{A} = [\mathbf{a}_1, \ldots, \mathbf{a}_R] \geq 0, \mathbf{B} \geq 0, \tag{61}$$

$$\mathbf{a}_r = \boldsymbol{\phi}(u|\theta_r) \in [0,1]^{N_i}, \tag{62}$$

$$\|\mathbf{a}_r\|_\infty = 1 \text{ for } r = 1, 2, \ldots, R. \tag{63}$$

For other works using DIP unsupervised method, the reader can refer to [197–199].

### 5.5. Discussion

Because the measuring model of PET is the most complex (Poisson noise and ill-posed measurement matrix), it is hard to design $P_{\mathrm{dc}}$ and different unrolling methods. Therefore, image domain is usually considered in PET reconstruction and it is often regarded as a denoising problem using popular U-Net, which is somehow similar to CT reconstruction.

## 6. Discussion and Future Directions

We have reviewed many works of the deep learning application in CS, CT, MRI and PET reconstruction. Though they are different in details, these works hold a common character, satisfying framework $\mathcal{F}$ which is described in Section 1. In general, most neural networks play the role of $P_{\mathrm{image}}$. Therefore, the reconstruction framework is the same to traditional methods.

We may ask: what is the advantages of deep learning? In the framework $\mathcal{F}$, $P_{\mathrm{image}}$ is the key part, because in most cases, the $P_{\mathrm{dc}}$ is easy to derive. Thus, the performance of a reconstruction algorithm usually depends on the design of $P_{\mathrm{image}}$. For traditional methods, $P_{\mathrm{image}}$ is derived from a hand-designed image prior distribution or regularization terms. Even if some method are similar to data-driven methods such as dictionary learning, the L1 norm and the form of linear transform are determined in advance. The drawback of the hand-designed model is that it may be insufficient or inaccurate to depict the real prior distribution of signals. However, deep learning holds two advantages that make it

successful. Firstly, it is data-driven. If a large dataset is available, the model can directly utilize the distribution information hidden in training data. Secondly, it allow researchers to design more complex and flexible model to better represent the image prior distribution.

However, deep learning also has a disadvantage that has not been solved well. In [200], three tests were used to inspect the stability of deep learning. Several popular models are compared to traditional methods. The results show that there is some instability problem in deep learning, while there isstrong stability for traditional methods. The lack of training data may be another handicap for medical applications.

Nevertheless, deep learning provides a powerful tool which can be used to learn prior information from data. More specifically, there are three types of models proposed in current research. Neural networks are used to depict $P_{image}$, $p(\mathbf{x})$ or $M_{image}$ directly. Usually, a CNN denoising-like model is used to represent $P_{image}$, the projection operator (see Sections 2.3, 3.3, 4.3 and 5.2). In the MAP method, the network is exploited to compute the prior distribution $p(\mathbf{x})$ (see Sections 2.5 and 4.5). Generative models are utilized to depict image manifold $M_{image}$ (see Sections 2.4, 4.4, 4.6, 5.3 and 5.4).

As for future research, how to design more efficient network is a clear direction. We have seen three types of deep learning model. Which one is the best? What is the relationship between different models, $P_{image}$, $M_{image}$ and $p(\mathbf{x})$. These questions have not been answered. Actually, $P_{image}$, $M_{image}$ and $p(\mathbf{x})$ are different facets of one thing, the image prior. Combining them all may a feasible way to design novel networks. The breakthrough of deep learning theory ought to be helpful. It may tell us how a network plays the role of $P_{image}$ or how it can represent a complex manifold. It can also provide new ideas to train the model. In addition, the statistical properties of image or signal prior distribution can inspire researchers to design more feasible and robust network architectures. For example, the property of multi-scale has been considered in many works. Few shot learning, robustness of networks and computation efficiency are also worthy of attention. Besides, the theoretical properties of deep learning reconstruction methods, including the existence and uniqueness of solution and the convergence of algorithm, are important research directions in the future. Another important issue discussed less in this review is the measurement noise and artifacts, which will lead to noisy images in real life. It is necessary to alleviate the effect of noise and artifacts. One of methods is pre–Processing of these noisy images, for example, denoising using 1st and 2nd generation wavelets [201]. Readers can refer to [202] for more works about it. Security [203] and privacy-perserving problem [204] are also important in image reconstruction, especially in medical image reconstruction tasks. However, they have not been studied deeply. In addition, compressed sensing, or inverse problem also exists in the area of surveillance [205], medical [206], agriculture [207], speech [208] and telecommunications. Our proposed framework may be helpful to inspire researchers to improve their works.

## 7. Conclusions

Deep learning has been proved to be successful in CS reconstruction. In this paper, we review some works on it and its medical applications using deep learning methods. A framework $\mathcal{F}$ is derived to better understand these approaches. We define two projection operators toward image prior and data consistency, respectively, and any reconstruction algorithm can be decomposed to the two parts. Based on it, several categories are analyzed and relationship between them is built. It also helps us to connect deep learning methods to traditional iterative algorithms. Our analysis illustrates that the key to solve CS problem and its medical applications is how to depict the image prior to this. We hope that the proposed framework and our observation may provide a new perspective to improve the current work.

**Author Contributions:** Conceptualization, Y.X.; investigation, Y.X.; writing—original draft preparation, Y.X.; writing—review and editing, Q.L.; visualization, Y.X.; supervision, Q.L.; project administration, Q.L. All authors have read and agreed to the published version of the manuscript.

**Funding:** This research received no external funding.

**Conflicts of Interest:** The authors declare no conflict of interest.

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
