# Peer review of "A Review of Deep Learning Methods for Compressed Sensing Image Reconstruction and Its Medical Applications"

_electronics, doi:10.3390/electronics11040586_

Round 1
Reviewer 1 Report
The article discusses various Deep Learning Methods for Compressed Sensing
Image Reconstruction and Its Medical Applications. The study is well elaborated and discusses many significant advancements. It is well-drafted document with explanations. my observations are:
- The abstract of the article can be enhanced with a focus on outcomes and the importance of this study.
- The contribution section needs to add to the introduction.
- The authors can discuss section/area wise articles in the paper.
- Put a discussion section after a particular section of articles.
- I suggest including the following relevant articles in this study. as this is review article so i suggest to cite these articles
- "Secure video communication using firefly optimization and visual cryptography", "A bio-inspired privacy-preserving framework for healthcare systems"
Author Response
We would like to thank you for your valuable comments.
Reviewer 2 Report
This is a comprehensive and detailed review article. It is a great addition to the literature.
Author Response

(The authors gave the same response as above.)

Reviewer 3 Report
Authors should address the following major revision.
1) Add the importance of deep learning with several applications such as surveillance, medical, agriculture and named few more. For each application, the following work can be considered: i) Human Action Recognition: A Paradigm of Best Deep Learning Features Selection and Serial based Extended Fusion; ii) A Rapid Artificial Intelligence-based Computer-Aided Diagnosis System for COVID19 Classification from CT Images; Cucumber Leaf Diseases Recognition Using Multi Level Deep Entropy-ELM Feature Selection
2) "Instead, some pre-fixed parameters or functions in traditional algorithms are learned from training data. Generally speaking, through loss function and back-propagation method, these algorithms can be regarded as a trainable network."- refine this statement more precisely and add some references.
3) What represent \Delta and K in equation 8? add the detail of all symbols in the manuscript or it is better to add a nomenclature table.
4) What is \phi in Eq. 9? why adding too many equations without any reason?
5) Deep learning methods needs to be better explaination.
Author Response
For Point 1, we cited some papers in Section 6 to show the importance of deep learning in several other applications such as surveillance, medical and agriculture. For Point 2, we modified the statement and explained it more precisely and added some references. As for Point 3 and 4, we added some explanations for the notations. Because the methods reviewed in this article vary widely, it is hard to use consistent notations and add a nomenclature table. We added those equations in our article in order to illustrate the corresponding method in a concise manner. For the last point, we add a paragraph to the introduction to explain deep learning methods.
Round 2
Reviewer 3 Report
All my comments are successfully addressed by authors. I recommend to accept this work.
Author Response

(The authors gave the same response as above.)
